# Peer review of "Error-correcting codes for fermionic quantum simulation"

_SciPost Physics_

## Round 1 · Referee Report · Anonymous (Referee 1) · 2023-11-10

Strengths

  1. The authors proposed a way to generate higher distance codes

  2. Clear presentation of the methods and algorithms

  3. Pedagogical discussion of 2D bosonization, the stabilizer code formalism, and the Pauli module representation via Laurent polynomials.

Weaknesses

  1. The authors present a rather straightforward generalisation of previously known models, using standard techniques.

  2. There is no discussion of implementation of these models on NISQ hardware.

Report

The authors present a generalisation of 2D bosonization [24] to codes with larger distances. They explain how to use the Laurent polynomials method to perform these extensions in a general way.

This paper should be of interest to experts in the quantum computations/quantum information communities, and I recommend it for publication in SciPost.

Requested changes

I have noticed some spelling errors, which should be corrected.

  • validity: good
  • significance: ok
  • originality: ok
  • clarity: high
  • formatting: excellent
  • grammar: excellent

Author:  Yu-An Chen  on 2023-11-23  [id 4142]

(in reply to Report 1 on 2023-11-10)
Category:
remark
correction
suggestion for further work

We are grateful for the referee's insightful summary and constructive feedback. We have revised our manuscript to address the issues raised.

1. We have proofread the text and equations, correcting any typos present.
2. The description of the symplectic group and automorphisms has been elaborated upon. Section 3.1 has been comprehensively rewritten, with the complete list of generators for the symplectic group added to Appendix B.
3. We have refined the introduction to enhance clarity.
4. Additional references related to the topic have been incorporated into the introduction.
5. The manuscript's format has been updated to conform to the guidelines provided by SciPost Physics.

We now turn to address the concerns highlighted by the referee.

1. The referee noted that our work appears to be a straightforward generalization of existing models employing standard techniques. While we acknowledge that both the models and techniques have been previously established, as mentioned at the bottom of page 3, the innovation of our work lies in the novel application of the "Laurent polynomial" analytical tool to a numerical search algorithm. This synergistic approach could solve other optimization problems in quantum coding.

2. Implementing these models on NISQ hardware is a crucial consideration for the practical application of our proposals. The intricate discussion of implementing our fermion-to-qubit mappings highly depends on the specific hardware platforms and requires in-depth investigation. We intend to explore this in subsequent research. Our current manuscript compares the Pauli weights of standard Hamiltonian terms relative to other established approaches in Table I. This comparison offers insights into the practicality of implementing these mappings in experimental settings.

Anonymous on 2023-11-29  [id 4157]

(in reply to Yu-An Chen on 2023-11-23 [id 4142])

I am happy with the response, and recommend the paper for publication.

---

## Editorial Decision

resubmitted